# Identifying Urban Functional Regions from High-Resolution Satellite Images Using a Context-Aware Segmentation Network

**Wufan Zhao** [1] , **Mengmeng Li** [2,*] , **Cai Wu** [1] , **Wen Zhou** [1] and **Guozhong Chu** [2]

1   Faculty of Geo-Information Science and Earth Observation (ITC), University of Twente,
    7500 AE Enschede, The Netherlands
2   Key Lab of Spatial Data Mining & Information Sharing of Ministry of Education, Academy of Digital
    China (Fujian), Fuzhou University, Fuzhou 350108, China
*   Correspondence: mli@fzu.edu.cn

**Abstract:** The automatic identification of urban functional regions (UFRs) is crucial for urban planning and management. A key issue involved in URF classification is to properly determine the basic functional units, for which popular practices are usually based upon existing land use boundaries or road networks. Such practices suffer from the unavailability of existing datasets, leading to difficulty in large-scale mapping. To deal with this problem, this paper presents a method to automatically obtain functional units for URF classification using high-resolution remote sensing images. We develop a context-aware segmentation network to simultaneously extract buildings and road networks from remote sensing images. The extracted road networks are used for partitioning functional units, upon which five main building types are distinguished considering building height, morphology, and geometry. Finally, the UFRs are classified according to the distribution of building types. We conducted experiments using a GaoFen-2 satellite image with a spatial resolution of 0.8 m acquired in Fuzhou, China. Experimental results showed that the proposed segmentation network performed better than other convolutional neural network segmentation methods (i.e., PSPNet, Deeplabv3+, DANet, and JointNet), with an increase of F1-score up to 1.37% and 1.19% for road and building extraction, respectively. Results also showed that the residential regions, accounting for most of the urban areas, identified by the proposed method had a user accuracy of 94%, implying the promise of the proposed method for deriving the spatial units and the types of urban functional regions.

**Keywords:** urban function regions; high resolution satellite images; building extraction; road extraction; context-aware semantic segmentation

## 1. Introduction

Most people live in urban areas, where the world is undergoing rapid urbanization, particularly in developing countries, posing the urgent need for sustainable urban development [1]. Spatial information such as urban land use and function, which refers to the use of the land, is thus essential for effective urban planning and management [2–7].

The urban function prioritizes the socioeconomic functional aspects of cities. Dubrova et al. [8] demonstrated a method to divide land use by function and called this functional zoning or functional city zoning. This topic has since been studied by many scholars [3,9–11]. The function of an area is defined according to its purpose, e.g., for living, working, and production, for human beings, corresponding to land use types of residential, commercial, and industrial (Urban Functions (coolgeography.co.uk, accessed on 5 June 2022)). The urban function is defined from the perspective of urban services, which have been provided in a particular metropolitan area and are highly related to urban land use. For example, according to the New York City Department of City Planning (NYCDCP) [12], land use mapping of buildings can be classified into "one and two-family buildings, multi-family walk-up buildings, multi-family elevator buildings". From the

urban function description, these three land-use categories reflect the same urban function, i.e., residential. Therefore, the relationship between land use and urban function is that urban land use can often reflect urban function. The social-economical attributes reflected by urban function are usually more abstract and coarser than many detailed land use categories. For an urban area, describing and mapping land according to its spatial attributes can be called land cover mapping [2,13], while according to its social-economical attributes is land use mapping. From this point, mapping land according to its urban function is also a kind of urban land use mapping.

Traditional methods of urban function mapping rely significantly on time-consuming and expensive labor-intensive land surveying. To alleviate the problem, recent work proposes numerous methods covering machine learning and deep learning [2,14–16]. From a processing strategy standpoint, "top-down" (division or segmentation) and "bottom-up" (aggregation) are two often utilized ways for exploring basic functional regions [3]. Among those studies, the extraction of urban function information based on remote sensing images have been developed rapidly, thanks to the development of high-resolution optical satellite remote sensing technology (e.g., the launching of Geo-Eye, WorldView, and Pleiades) [17–22]. In addition to that, social sensing data-driven solutions are emerging in response to the growing availability of social sensing big data (e.g., geotagged social media, vehicle trajectories and public transport data, and street view images) [11,23–25]. The increase of these data enables a more comprehensive depiction of the diverse urban functions they reveal. However, there are difficulties in fusing multiple types of data, and the final classification results depend more on the reliability of social sensing data. In addition, compared with remote sensing images, the crowd-source data-based method has efficiency drawbacks in large-scale mapping.

Effective UFR mapping can be achieved by extracting and aggregating fine-grained information in remote sensing images. Therefore, one key factor is the determination of logical units and elements. Buildings are meta structures that house a variety of human activities. Building types data are also frequently used to measure human activity and further characterize urban functions [26]. However, urban areas with complex building types, shapes, sizes, and diverse functional uses make it difficult to identify different building types. Existing methods classify building types by analyzing the shape, morphology, spatial relationships, and other high-level image features of building objects extracted from images [26]. The studies prove that combining height information can effectively improve the classification of building types [14].

Moreover, basic geographic units are combinations of geographic elements at a specific scale, for which different scales provide distinct views of a city. Most studies choose grids, blocks, or cadastral boundary partition units for UFR mapping. However, rigid administrative boundaries are distracted from specific urban economic activities or basic divisions. In light of this, individuals prefer to interact with urban areas through road blocks. A road block is planned and constructed by humans and refers to an area with homogeneity regarding physical, social, or functional characteristics. Road blocks can be seen as natural spatial units linking between the macro-level (e.g., urban function area) and the micro-level (e.g., building types). The local area defined by the road block can also capture slight changes in the street layout.

Building and road information is essential to the identification of UFRs. Traditional research mainly collects vector building and road data by manual labeling or using open-source databases such as Open Street Map (OSM) [5]. However, manual labeling is often time-consuming, while open-source datasets have limited coverage and cannot be updated on time. Therefore, it is crucial to develop a method to automate the extraction of buildings and roads from remote sensing images. Traditional building and road extraction studies focus on using pixel- or object-based methods [27]. Convolutional neural networks (CNNs) have recently acquired prominence owing to their overwhelming effectiveness and strong generalizability for many applications. Since the introduction of Fully Convolutional Networks (FCNs) [28], semantic segmentation neural networks have developed several

distinctive frame-works and modules by integrating dilated convolution, multi-scale learning, and attention module [29–32]. However, there were issues with remote sensing image segmentation, because the building and road segmentation context was uneven, and the full extent of information was not sufficiently exploited. Due to the complexity and diversity of buildings and roads, extracting multi-level image features with strong representation is challenging when using a unified deep neural network for building and road extraction. There are several distinctions between road and building elements regarding image attributes. Buildings with blocky and large sizes require large receptive fields to cover context information. Roads, linear targets, require the network to focus on segmentation surfaces or connectivity between closed pixels. Thus, it is crucial to investigate effective contextual representation strategies that explore the relationship between a position and its contexts in remote sensing images (Figure 1). Some seminal works have been conducted using deep learning, such as atrous spatial pyramid pooling (ASPP) [30] and pyramid pooling module (PPM) [33], which explore multi-scale contexts by considering the relationships between a position and its contextual positions. The representations of the contextual positions are then combined using a weighting system that favors representations that are similar to one another. Inspired by these studies, we extend the representation of a single pixel into a region representation of the associated class.

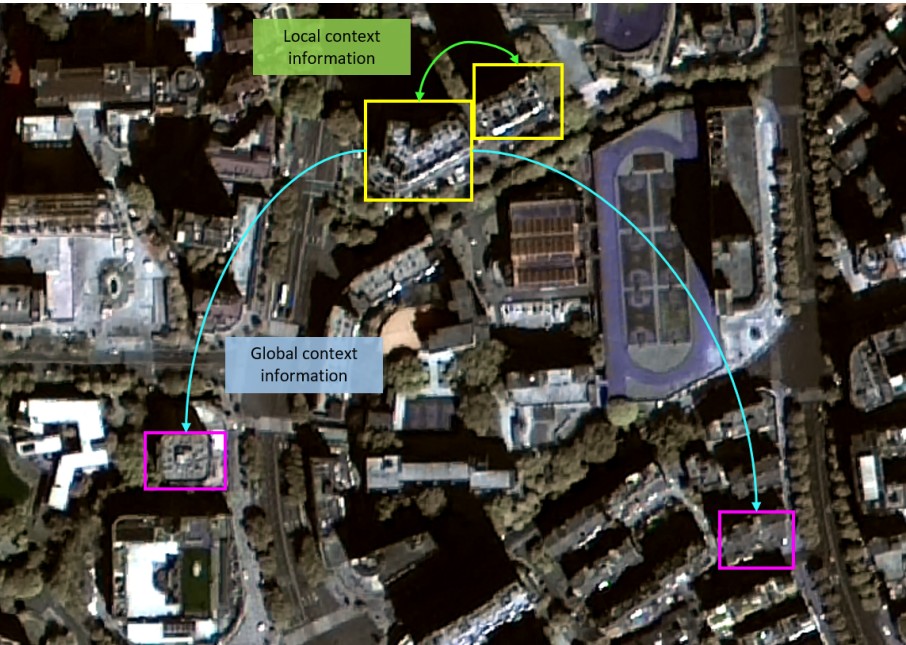

**Figure 1.** Illustration of contextual information in a remote sensing image.

The novelty of this study is as follows:

- Developing an efficient context-aware building and road segmentation network based upon a high-resolution feature conserving network and a region context module (RCM) for feature extraction. In addition, an affinity loss derived from the Region Affinity Map (RAM) is used to promote the network's training.
- Presenting an end-to-end method for directly extracting urban function regions using high-resolution satellite images, based upon image-derived functional units (i.e., road blocks) and multi-scale building features.

The remainder of this paper is structured as follows: the study area and dataset are described in Section 2, and the proposed UFR classification method based on remote sensing images is shown in Section 3. Section 4 gives experimental analysis, followed by discussion in Section 5 and conclusions in Section 6.

## 2. Study Area and Dataset

The study area is located in Fuzhou City, the capital of Fujian Province, China (Figure 2). Fuzhou, sitting along the Chinese southeast coast, is a renowned cultural city with a more than 2000-year history and diverse architectural styles. Like many other Chinese cities, it has also been undergoing fast urbanization, leading to an increase in new built-up regions. The study area covers the core urban area of Fuzhou City, including Gulou, Taijiang, Jin'an, and Cangshan District, accommodating most of the social and economic activities in this city. Here, homogenous functional regions are usually packed with various building types, formulating densely adjacent historic structures, regularly aligned residential structures, and irregular commercial buildings.

For this study, we acquired a high-resolution remote sensing image from the Gaofen-2 (GF-2) satellite on 18 February 2020. The study image includes one panchromatic band with a spatial resolution of 0.76 m and four multi-spectral bands with a resolution of 3.2 m. A preprocessing was conducted for this image, including image fusion, image resampling, and image cropping. The processed image has a size of 12,943 × 9673 pixels and a spatial resolution of 0.8 m. Moreover, we selected two 3 km ∼ 4 km subsets inside the GF-2 image for collecting training and validation samples. For experiment convenience, we further cropped the image into 256 × 256 patches. We applied the same preprocessing strategy to all comparison experiments.

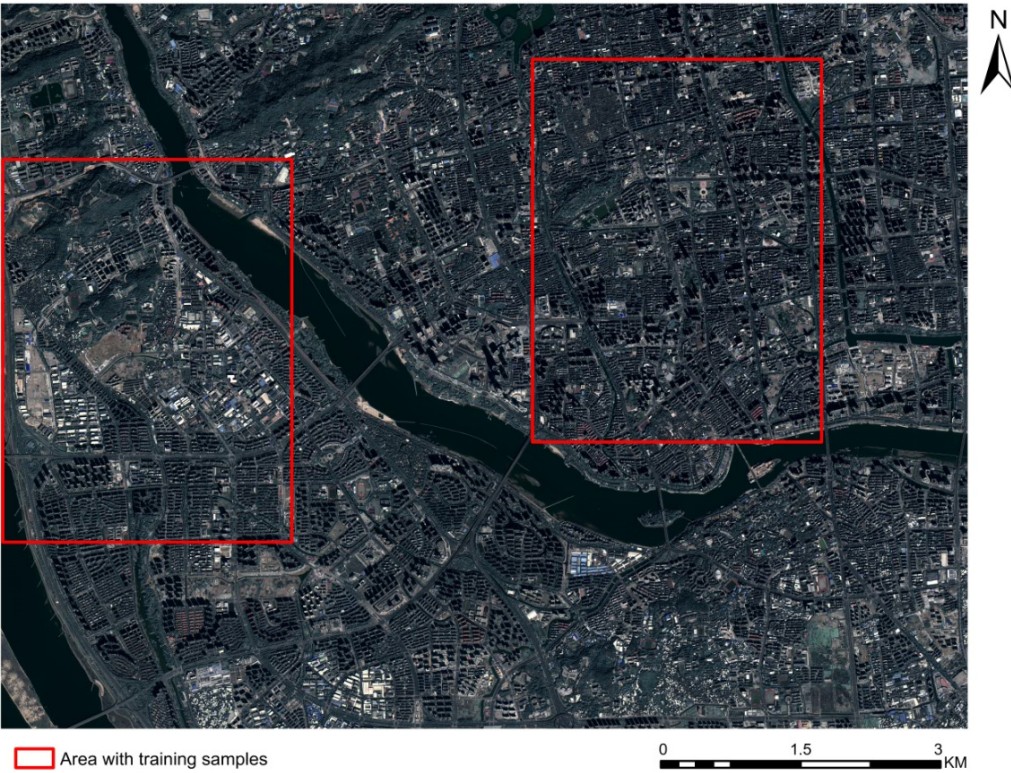

**Figure 2.** Overview of the study area. Red boxes refer to the subset areas of training and validation samples.

## 3. Methods

The proposed workflow is illustrated in Figure 3. First, we build a context-aware deep learning network for building and road network extraction from remote sensing images. The extracted road networks are then used to derive functional units (i.e., road blocks), while the extracted buildings are further classified into different types based upon a series of multi-scale features regarding the height, spatial structure, and geometry of building objects. Next, we distinguish different URFs based upon image-derived functional units and classified building types.

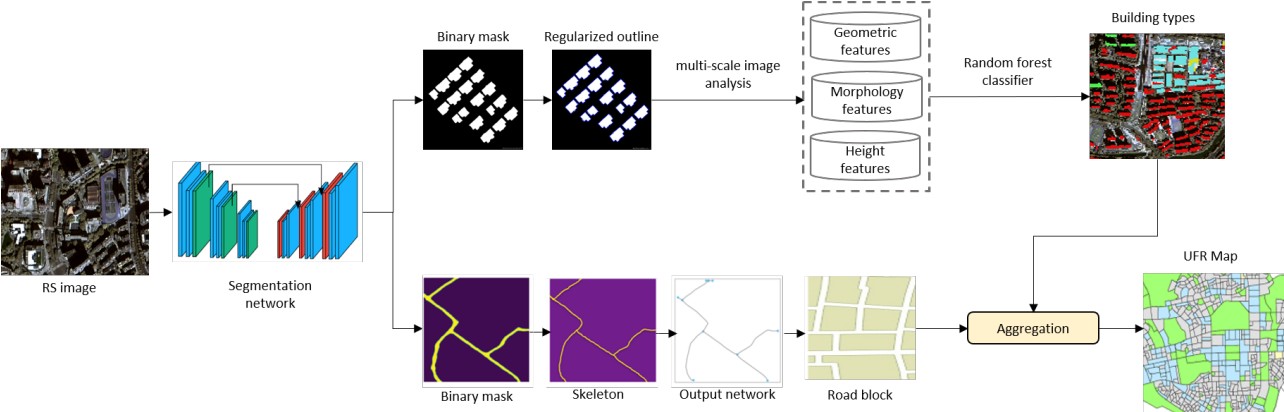

**Figure 3.** Overview of the proposed workflow. It inputs a remote sensing image to extract buildings and road networks, goes to building type classification based upon multi-scale image analysis, then ends with UFR identification according to the statistical characteristics of building types within the spatial units defined by the road network.

### 3.1. Building and Road Extraction

We develop an end-to-end trainable network for building and road extraction based on RCM (i.e., region context module). In this section, we first elaborate on the overall structure of our network (see Figure 4). Our fully convolutional network is composed of a backbone network and an RCM. The RCM models the relationship between the pixels belonging to the same category (intra-context) and those belonging to distinct categories (inter-context). It improves the applicability of the network for learning different features of ground objects (i.e., buildings or roads). Finally, we use an affinity loss to supervise the training of our network.

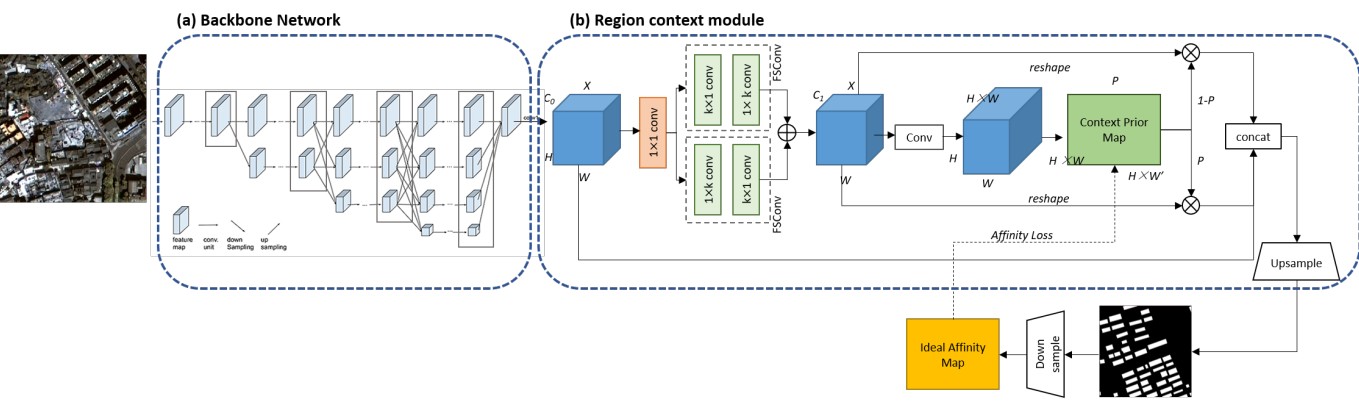

**Figure 4.** Overview of the proposed building and road extraction network. It is composed of (**a**) HRNet backbone network, and (**b**) a region context module (RCM). The RCM uses aggregation module output to learn a region affinity map that gathers contextual information surrounding each pixel. The RCM output feature map is then upsampled for prediction. (Notation: *Conv* standard convolution, *FSConv* fully separable convolution).

### 3.1.1. Semantic Segmentation Network for Building and Road Extraction

a.　Overall architecture

Semantic image segmentation is the task of assigning to each pixel the class of its enclosing object or region as its label, thereby creating a segmentation mask. Figure 4 illustrates the overall structure of the proposed network, which is an encoder-decoder neural network. The segmentation network uses RCM to explicitly distinguish the difference between different contextual relationships. Here, we choose HRNet as our backbone network, which retains high-resolution characteristics by supplying *n* stages, matching *n*

branches, and *n* resolutions [34]. HRNet begins with a high-resolution sub-network and gradually adds low-resolution sub-networks to construct different branches while linking the multi-resolution sub-networks concurrently. To guarantee the model's accuracy, we apply a lightweight version of HRNet, which adopts the parallel-branch ResNet as the backbone. The number of each stage is reduced to one, and the minimum residual units are kept in each branch of the same stage. After that, the spatial information is aggregated using two asymmetric separable convolutions. The output has the same channels as the input features. The features are taken as input to the RCM, which we described in detail below.

b.     Region context module

Following the backbone network, we take an input feature $X$ with the size $H \times W \times C$. An aggregation module is used to efficiently aggregate certain spatial information for the RCM. The RCM creates a prior context map based on the aggregated information to capture intra- and inter-class contexts. The aggregation module applies two separate convolutions, yielding two new feature maps $X_1$ and $X_2$, with the shape $H \times W \times C'$. Then, we multiply $X_1$ by $X_2$ to obtain the attention map, where $X_1$ is reshaped as $HW \times C'$ and $X_2$ as $C' \times HW$. Next, the attention map is fed into a Sigmoid layer, the output of which is an affinity map for projected regions, denoted by $A_1$. $A_1$ compares the attributes of pixels without regarding their spatial connection. To compensate for its shortcomings, the network creates an auxiliary prediction $X_p$ to obtain a pseudo label $Y_p$ from $X_p$. We repeat the same operation for $Y_p$ for the ground truth to obtain the predicted area affinity map $R_1$. Then, we dot products $R_1$ and $A_1$ to encode region context information in $A_1$. The outcome is $R_A$, monitored by the region loss described in the following subsection.

After obtaining the RAM $R_A$, the region context feature $Y$ is defined as $Y = R_A \otimes X$, with $X$ reshaped to the $HW \times C$ size. The RAM is used to adaptively select intra-region pixels as the regional context for each pixel in the feature map. Finally, we concatenate the initial feature with the area context feature to produce $F = Concat(X, Y)$. This method is analogous to residual connection, which preserves the original data while maintains area context.

c.     Affinity loss

In addition, an affinity loss is used to regularize the network to describe the link between categories explicitly. Intra-context and inter-context pixels are considered for each pixel, resulting in a loss for each pixel in an image (inter-context). We begin by generating an affinity map from an input image $I$ and the ground truth $L$. To begin, the downsampled ground truth $\tilde{L}$ is encoded using one-hot encoding. The size of the ground truth, $\hat{L}$, is equal to $H \times W \times C$, where $C$ is the number of classes. Each entity in $\hat{L}$ has a single high value (1), and all the rest are low values (0). We use $A = \hat{L}\hat{L}^\top$ to build the affinity map $A$, where $\{a_n \in A, n \in [1, N^2]\}$ is the intended affinity map of size $N \times N$, containing information of pixels with the same category. Together with a prior map $P$, $\{p_n \in P, n \in [1, N^2]\}$, we consider the intra-class and inter-class pixels as two distinct wholes to encode their relationships respectively. To this end, the binary cross entropy loss function is represented as:

$$\mathcal{L}_p = -\frac{1}{N^2} \sum_{n=1}^{N^2} (a_n \log p_n + (1 - a_n) \log(1 - p_n)) - \frac{1}{N} \sum_{j=1}^{N} \left( \mathcal{T}_j^p + \mathcal{T}_j^r + \mathcal{T}_j^s \right), \qquad (1)$$

where global term $\mathcal{T}_j^p$, $\mathcal{T}_j^r$, and $\mathcal{T}_j^s$ represent the intra-class predictive value (precision), true intra-class rate (recall), and true inter-class rate (specificity) at *j*th row of $P$, respectively.

Under the supervision of an affinity loss, we create a point-wise context prior map (see Figure 5). The affinity loss generates an affinity map that identifies the pixels that belong to the same category to monitor the RCM's learning. We can retrieve the intra- and inter-prior ($P$) priors using the RCM $(1 - P)$. The original feature map is rebuilt to have a dimension of $N \times C_1$, where $N = H \times W$. To capture intra- and inter-class context,

we perform matrix multiplication on the redesigned feature map using $P$ and $(1 - P)$. The prior map adaptively chooses intra-class pixels as the intra-class context for each pixel in the feature map. On another hand, the inverted prior map highlights inter-class pixels selectively as inter-class context. Finally, we input the context prior layer representation into the final convolutional layer to create a per-pixel prediction. We can infer each pixel's semantic relationship and scene structure using both contexts.

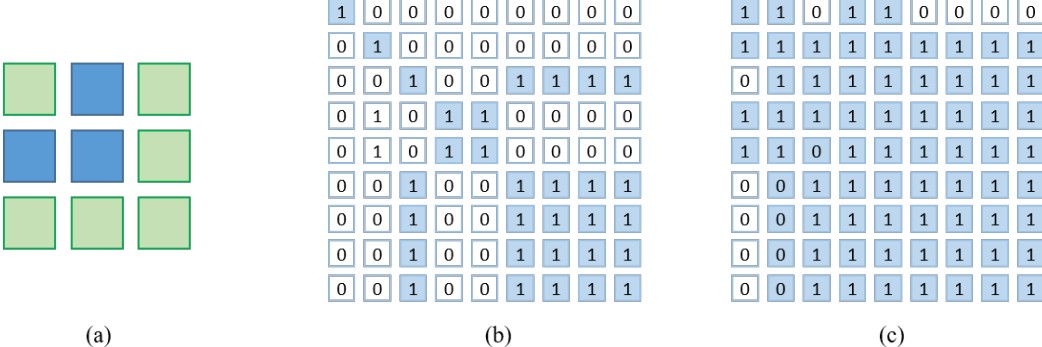

|       |       |       |
|:-----:|:-----:|:-----:|
| (a)   | (b)   | (c)   |

**Figure 5.** Illustration of the construction of the region affinity map (RAM). (**a**) shows the downsampled ground truth, with different colors representing distinct pixel categories. The RAM (**b**) represents the regional association between any two pixels in (**a**). A pixel's relationship degree is one if they are in the same region. The image on the right (**c**) represents the region affinity map after boundary relaxation. Compared to (**b**), we regard the boundary pixels in each region in (**a**) to be part of the same region as the surrounding pixels.

Another challenge in the building and road segmentation task from remote sensing imagery is boundary prediction, which is a problem for the preceding pseudo label. As a result, the final RAM assigns pixels on the boundary to the incorrect region, resulting in the wrong region context for the associated pixels. To address the concern above, we make a simple yet effective change. To put it another way, we see that, while the prediction of the pseudo label on the boundary is inaccurate, the pixels in this region are frequently misclassified to the category corresponding to the surrounding region. Therefore, we relax the pixels on the boundary, assuming that the region to which they belong is identical to the regions of all surrounding pixels.

The affinity loss regularizes prior context learning, while the cross entropy loss function provides segmentation supervision. The auxiliary loss, also a cross-entropy loss, is used in the final stage of the backbone network. The ultimate loss function is:

$$\mathcal{L} = \lambda_s \mathcal{L}_s + \lambda_a \mathcal{L}_a + \lambda_p \mathcal{L}_p \tag{2}$$

Here, $\mathcal{L}_s$, $\mathcal{L}_a$, and $\mathcal{L}_p$ denote the main segmentation loss, auxiliary loss, and affinity loss functions. Segmentation loss is compensated by $\lambda_s$ weight. An additional weight of $\lambda_a$ for auxiliary loss, and an affinity loss weight of $\lambda_p$. The above three parameters are set to 1, 0.4, and 1, respectively.

We compare our method with four other state-of-the-art methods: PSPNet [33], Deeplab V3+ [30], DANet [31], and Joint-Net [35]. Please see the comparison results in Section 3.1.2.

### 3.1.2. Post Processing

The results obtained from segmentation networks are usually blurry and imperfect. Subsequent spatial modeling and analysis often require vectorization and regularization of the preliminary results. Therefore, we applied post-processing methods to optimize the segmentation results for buildings and roads, respectively [36,37].

We applied a refined polygon regularization algorithm proposed by [36] for building regularization. It consists of two main steps: boundary extraction and polygonization.

In the classified building image, each connected component refers to a building object, for which the Marching Cubes technique is used to extract the building's boundaries. The extracted boundaries were further regularized by a coarse adjustment that corrects evident segmentation errors for the original polygons and a refinement adjustment that changes the direction of lines and nodes. Last, we use the Douglas–Peucker method [38] for the polygonization.

For road regularization, we refer to the work of [37]. The road skeleton mask was created by converting the probability map of road pixels to a binary image and applying skeletonization operation. The skeleton mask is represented using a graph data structure. Then, post-processing operations are carried out to straighten skeleton lines, fill undesired gaps, and remove unnecessary lines. We then use the Network Analysis tool provided by ArcGIS software to identify and construct road lines and intersections. Last, we construct a new polyline layer (i.e., road network) by spatially connecting the road lines to the intersections.

### 3.2. Building Types Classification and Urban Function Region Identification
#### 3.2.1. Building Feature Extraction

We further classify image-derived building objects into five building types, namely intensive buildings, middle and low-rise residential buildings, high-rise commercial and residential buildings, commercial buildings, and factory-type buildings (Table 1). For building type classification, we extract a number of features regarding building height, morphology, and geometry. Building height is estimated based upon the directional relationships between buildings and their shadows [39]. More specifically, a shadow object cast by its adjacent building can be identified. Its length is then computed by averaging the length of estimated lines parallel to the sun azimuth direction. In flat urban areas, we mainly consider two directional relationships, i.e., the sun and the satellite are positioned on the same and opposite sides of the target building object. Let $H$ be the height of a building object; for the former case, it can be estimated by

$$L_I = L_B - L_H = \frac{H}{\tan \alpha} - \frac{H}{\tan \beta}$$

$$H = L_I \times \frac{\tan \beta \times \tan \alpha}{(\tan \beta - \tan \alpha)} \tag{3}$$

where $L_I$ is the length of the shadow on the image; $L_B$ is the actual shadow length of the building; $L_H$ is the shadow length that cannot be displayed on the image due to building occlusion; $\alpha$ is the sun elevation angle; $\beta$ is the satellite elevation angle. When the sun and the satellite are on the opposite side of the target building, the building height $H$ is estimated as follows:

$$H = L_B \times \tan \alpha = L_I \times \tan \alpha \tag{4}$$

We also extract two morphological features based upon the profiles of morphological opening and closing operations [40]. More specifically, given an image object $R$, a series of opening and closing operations, $\gamma_R^{(i)}$ and $\oslash_R^{(i)}$, with structure element of different sizes (i.e., $i$) can be conducted, forming an opening profile ($OP$) and a closing profile ($CP$) [41],

$$OP(R) = \gamma_R^{(i)} \quad \forall i \in [0, n]$$

$$CP(R) = \oslash_R^{(i)} \quad \forall i \in [0, n] \tag{5}$$

where $n$ denotes the number of opening or closed operations. The means of the $OP$ and $CP$ are used as morphological features in this study.

In addition, we extract nine geometric features of building objects, i.e., area, perimeter, roundness, tightness, aspect ratio, boundary index, shape index, rectangular similarity, and elliptical similarity, for building type classification.

**Table 1.** Demonstration of building types.

| Building Types | Sample Image | Description |
|---|---|---|
| Intensive building (B1) |  | Mainly refers to a group of closely adjacent buildings. Usually, the open space and vegetation cover density around this type of building is low. |
| Middle and low-rise residential buildings (B2) |  | Mainly refers to middle and low-rise residential buildings, which usually have similar shapes and sizes in the image, with more regular spatial arrangement and higher density of vegetation cover and open space than intensive buildings. |
| High-rise commercial and residential buildings (B3) |  | Mainly refers to high-rise residential and commercial buildings, which have similar visual effects to middle and low-rise residential buildings in images, but with more floors and often adjacent to complete regular shadows. |
| Commercial buildings (B4) |  | Mainly refers to composite buildings used for commercial services or offices, which tend to have irregular shapes, staggered heights, larger areas than residential buildings, and more open spaces. |
| Factory-type buildings (B5) |  | Mainly refers to industrial or warehouse storage factories with a larger area than residential buildings. Compared to commercial buildings, factory-type buildings have more regular shapes and homogeneous spectral responses. |

### 3.2.2. Building Type Classification

Multi-scale image analysis has been a popular strategy for object classification [40]. In this study, we also adopt the idea of multi-scale image analysis to extract building features at multiple scales, and subsequently to classify building. We first segment the image-derived buildings into image objects at different scales, forming a hierarchical segmentation representation. Let $f_R^l$ be the feature vector of the image object $R$ at the $l$ segmentation level, i.e., $f_R^l = \left\{ f_{R,1}^l, \cdots, f_{R,N}^l \right\}$, with $N$ image features. We can then formulate a new feature vector $F_R^l$ by concatenating $f_R^l$ at multiple levels, i.e., $F_R^l = \left\{ f_{R,1}^l, \cdots, f_{R,N}^l \right\}$ with $L$ levels. In this study, we represent building objects at three levels, i.e., $L = 3$, corresponding to setting scale values to 10, 20, and 30 for the multi-scale image segmentation of image-derived buildings. Based upon multi-scale image features, we use a random forest (RF) [42] to classify building objects into five types.

### 3.2.3. Urban Function Region Identification

We refer to previous related studies for designing a URF classification system [3,5,43]. Considering the sizes of functional units and the classified building types, we focus on distinguishing coarse-grained UFR classes, namely residential, commercial, industrial, and other uses. The functional units are obtained by partitioning the image-derived road networks. Then, we counted the number of different types of building objects per block. Two knowledgeable volunteers manually named each block by referring to the street view of the Baidu map and points of interest (POI) data. The division basis of functional area

types is shown in Table 2. This study disregards the functional segmentation of complicated mixed-use areas.

**Table 2.** Decision rules for urban function regions based upon different building types (B1–B5).

| UFR Types | Description |
|---|---|
| Residential region (R1) | The absence of B3 or B5 in the block with more than 50% of B1 and B2. |
| Commercial region (R2) | The presence of B3 in the block, or the proportion of B4 exceeds 50%. |
| Industrial region (R3) | The presence of B5 in the block. |
| Other uses (R4) | Other blocks, including green areas, water bodies, etc. |

*3.3. Evaluation Metrics*

This research evaluates segmentation accuracy using overall accuracy (*OA*), intersection ratio (*IoU*), recall, accuracy (*Precision*), and *F*1 *Score*.

$$OA = \frac{TP + TN}{TP + FN + FP + TN} \tag{6}$$

$$IoU = \frac{TP}{TP + FN + FP} \tag{7}$$

$$\text{Recall} = \frac{TP}{TP + FN} \tag{8}$$

$$Precision = \frac{TP}{TP + FP} \tag{9}$$

$$F1\ Score = \frac{2 \times \text{Precision} \times \text{Recall}}{\text{Precision} + \text{Recall}} \tag{10}$$

where $TP$ (True Positive) is a correctly classified class; $FP$ (False Positive) is a negative class misclassified as positive; $TN$ (True Negative) is a correctly classified negative class; $FN$ (False Negative) is a positive class misclassified as negative. Additionally, the confusion matrix is used to evaluate the attribute accuracy for the building type classification results and UFR categorization results.

We also apply the Frames Per Second (FPS) to estimate the different networks' inference speed on the same machine with identical settings.

*3.4. Implementation Details*

To avoid overfitting during the training phase, we performed mean subtraction, random horizontal flip, and random scaling with 0.5, 0.75, 1.0, 1.5, 1.75, and 2.0 on the input images. Ten thousand pieces of samples were obtained through data augmentation, and were further divided into a training set and a validation set according to the ratio of 4:14. In addition, an area of $3750 \times 3500$ pixels was selected to verify the accuracy of building extraction results.

The initial learning rate was set to 0.0001 during model training. We optimized the model using the stochastic gradient descent (SGD) method [44] with a momentum of 0.9, a weight decay of $10^{-4}$. The batch size was set to 16, and the number of model iterations was 50. When the validation set's accuracy and loss rate settle and the model converges, the best model was retained for extracting building and shadow information. On the last stage of the backbone network, we incorporated the auxiliary loss. In the aggregation module, we set the filter size of the completely separable convolution to 11.

## 4. Experiments and Results

### 4.1. Building and Road Extraction Results

4.1.1. Ablation Studies on Network Design

This section describes the ablation studies conducted to evaluate the efficacy of two critical components of the proposed segmentation model. The HRNet was used as the starting point, and each component was added one at a time. First, we examine the RCM's efficacy. As indicated in Table 3, the RCM enhances the F1 score and OA by 0.63% and 1.51%, respectively, over the baseline model. The improvement demonstrates the effectiveness of the proposed context prior. The result reveals that the affinity loss module also enhances the F1 score and OA by 1.10% and 0.82%, respectively, demonstrating that the two modules are complimentary.

**Table 3.** Ablation analysis of RCM and affinity loss on the Fuzhou testing dataset.

| RCM | Affinity Loss | OA | F1 Score | Gain |
|:---:|:---:|:---:|:---:|:---:|
| - | - | 89.12 | 80.52 | - |
| ✓ | - | 90.63 | 81.15 | 1.51/0.63 |
| ✓ | ✓ | 91.45 | 82.25 | 0.82/1.10 |

Designing generic detection networks for buildings and roads is challenging. The target's context information is conceptually related but scattered across a small area of the image's spatial space. Limited receptive fields cannot cover blocky and massive objects like buildings. Therefore, only networks with large receptive fields can identify them accurately. With a linear target, such as a road centerline, the segmentation surface or corner accuracy is reduced. As depicted in Figure 6a, when utilizing HRNet to achieve high-resolution feature retention and receptive field augmentation, the detection results will be impacted in the corner sections of buildings, while road extraction will be interrupted. By introducing the RCM, the learnt context prior extracts pixels that belong to the same class, whereas the inverted prior focuses on pixels that belong to distinct classes. The high level features of the deep network can be used to effectively increase the receptive field, while the low-level features are captured as information that can be reused for road extraction.

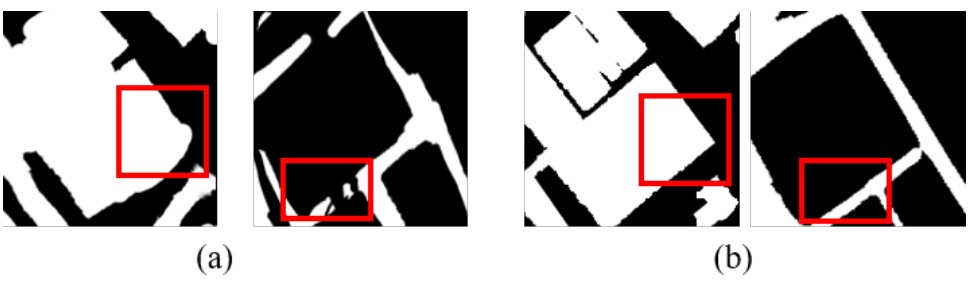

(a)  (b)

**Figure 6.** Comparing results between (**a**) the baseline method and (**b**) the baseline method plus RCM and Affinity loss. As shown in the red box, the results from the method with adding modules are better than the baseline method in terms of completeness and regularity of results.

4.1.2. Comparison with Other Methods

We compared our model with four existing methods (i.e., PSPNet, DeepLab v3+, DANet, and Joint-Net) on the Fuzhou testing dataset (Table 4 and Figure 7). Road and building extraction provide distinct challenges, i.e., road extraction encourages connectivity between short segments, while building extraction prefers reducing connectivity between objects and accommodating form diversity.

As shown in Figure 7, the majority of road networks extracted well in the regions with few roads and significant contrast. However, the road extraction becomes difficult when road topology is complex. In addition, our method fail to represent well for narrower

and tiny secondary roads, due to the background complexity. The roads extracted by our method show continuity advantages. Our method also works well for extracting small buildings, but has a relatively low extraction for large buildings. Noticeably, this low extraction happened for all methods. For medium-sized buildings, the proposed method shows better performance, providing clear extraction boundaries while maintaining the connectivity within the building. Overall, our method enables effective extraction of buildings and roads, producing more accurate results compared with the second-best method (i.e., DANet).

**Table 4.** Comparison of different networks on the Fuzhou testing dataset.

| Methods | Roads | | | | | Buildings | | | | |
|---|---|---|---|---|---|---|---|---|---|---|
| | OA | Recall | F1 Score | mIOU | FPS | OA | Recall | F1 Score | mIOU | FPS |
| PSPNet | 78.65 | 73.69 | 67.12 | 64.78 | 5.6 | 80.64 | 77.64 | 76.86 | 64.64 | 5.2 |
| DeepLab v3+ | 79.36 | 75.48 | 68.60 | 65.97 | 1.2 | 82.45 | 78.25 | 78.91 | 65.45 | 0.9 |
| DANet | 81.07 | 78.64 | 71.08 | 69.63 | 7.3 | 88.78 | 82.16 | 81.06 | 71.04 | 7.1 |
| Joint-Net | 80.64 | 77.12 | 70.16 | 68.74 | 11.4 | 86.80 | 80.76 | 80.17 | 69.87 | 11.0 |
| Our method | 82.87 | 80.36 | 72.45 | 70.29 | 13.9 | 91.45 | 84.36 | 82.25 | 72.68 | 13.5 |

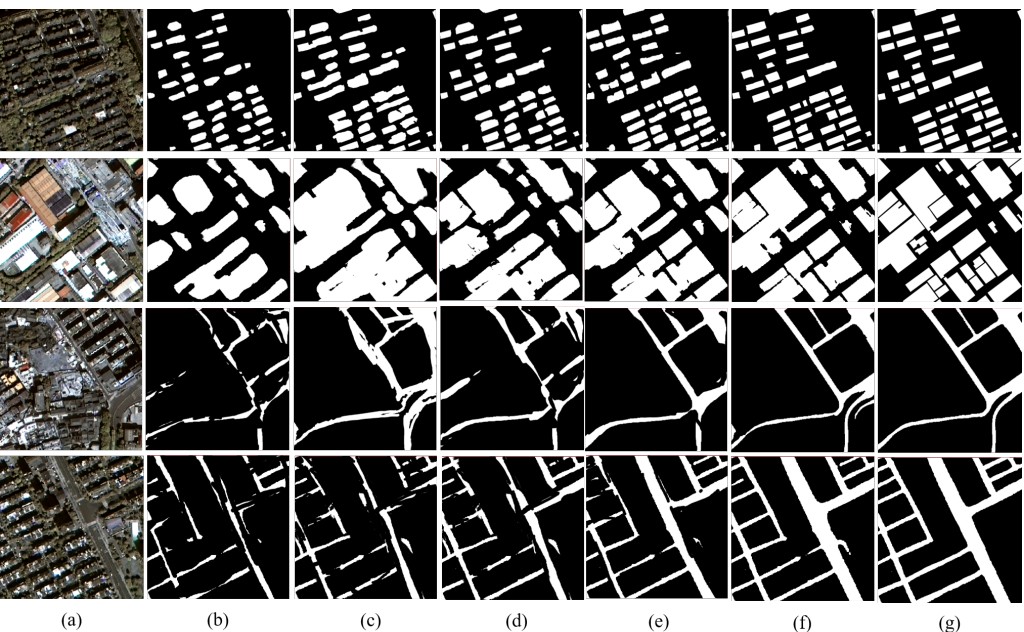

(a) (b) (c) (d) (e) (f) (g)

**Figure 7.** Results of building and road extraction. (**a**) Image patches. (**b**) PSPNet results. (**c**) DeepLab v3+ results. (**d**) DANet results. (**e**) Joint-Net results. (**f**) Our method results. (**g**) Labels.

More specifically, the majority of building objects in the study area can be extracted effectively. The errors are mainly caused by the mistakes in extracting artifacts (e.g., overpasses) and open spaces between adjacent buildings and omitting buildings under shadow occlusion. Unlike other methods, our network considers contextual dependencies before encoding the detected contextual relationship. We use the context prior layer to precisely record each pixel's intra- and inter-class contexts. Thus, the contextual module reduces the noise of objects and other irrelevant backgrounds to improve the discrimination. As a result, the proposed method had a lower likelihood of causing a discontinuity in the core region of extracted building and road objects than the other methods.

Table 4 shows that roads extracted by our method have an OA, recall, F1 score, and IoU of 82.87%, 80.36%, 72.45%, and 70.29%, and that extracted buildings of 91.45%, 84.36%, 82.25%, and 72.68%, respectively. Specifically, our method achieved 1.37% and 1.19% higher than the DANet among the existing methods on the F1 score. The results

demonstrated that our network performs better than other methods in all assessment measures for road extraction. Similarly, we see that the proposed method outperforms other comparison methods in all metrics of building extraction, which indicates that the results of building extraction are generally reliable and can effectively distinguish buildings from other objects. In addition, the FPS metrics show that our method provides superior accuracy and inference efficiency.

### 4.2. Building Types Classification Results

Figure 8 shows the 3D map of the building height estimation results of the study area. The estimated building heights in the study area range from 1.2 to 228 m. Among them, most of the buildings in the 0–50 m height range in the study area are old town buildings and low-rise residences; most of the buildings over 100 m in height are commercial buildings or international financial centers. The local details of image reveal that the distribution of low- and medium-rise buildings is concentrated, and that the shadow area of adjacent buildings is diminished. In contrast, the shadow of buildings adjacent to high-rise buildings is regular and complete, and the shadow area is more prominent. The height estimation results are consistent with the realities, showing that adjacent buildings tend to be of the same type and have little variation in height from one another.

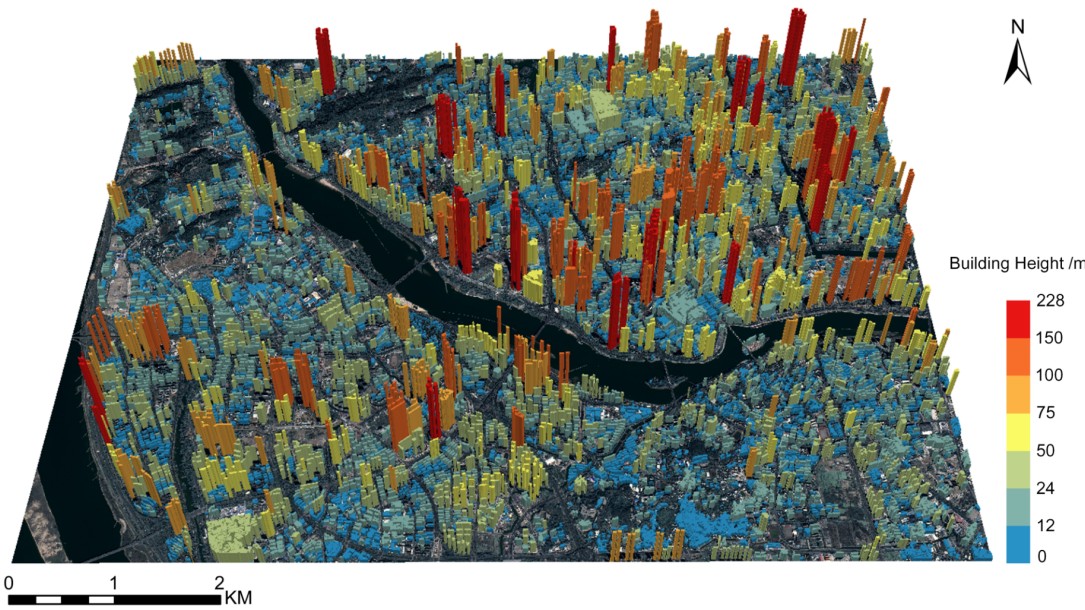

**Figure 8.** 3D demonstration of estimated building heights.

Through fieldwork and manual interpretation of high-resolution images, a total of 625 building objects were selected as classification sample data of the study area. The samples include 100 intensive buildings, 200 middle and low-rise residential buildings, 200 high-rise commercial and residential buildings, 25 commercial buildings, and 100 factory-type buildings. In addition, the validation samples were selected by stratified sampling for different types of buildings. 100 validation points were randomly chosen for intensive buildings, middle and low-rise residential buildings, high-rise commercial and residential buildings, and factory-type buildings. Besides, 20 validation points were selected for commercial buildings in the study area. The classification results in Figure 9 shows building type classification results. An average accuracy of 82.98%, and a kappa coefficient of 0.77 can be achieved. Among all building types, low-rise residential buildings have the highest accuracy of 94.04%, and commercial buildings have the lowest accuracy of 52.38%. Meanwhile, the height feature can effectively reduce the misclassification of residential buildings and improve the classification accuracy of middle and low-rise residential buildings and high-rise commercial and residential buildings.

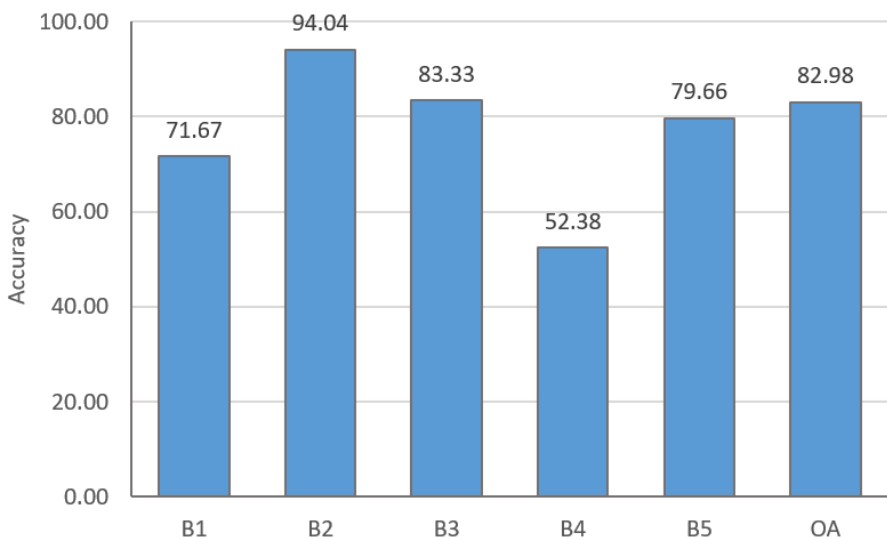

**Figure 9.** Classification accuracy of building types. B1~B5 stand for: intensive building, middle and low-rise residential buildings, high-rise commercial and residential buildings, commercial buildings and factory-type buildings, respectively.

Figure 10 shows the visualization results of the building type classification. Most building types are middle and low-rise residential buildings with a regular spatial arrangement. Each building has a similar shape and size, and is evenly distributed. Closely connected intensive buildings are mostly in old cities and shantytowns, and these types of buildings are adjacent to each other and distributed in blocks. In addition, high-rise commercial and residential buildings with residential and commercial applications are also prevalent in the study area. Most buildings of this type are distributed outside residential areas and on both sides of the rivers. There are fewer commercial buildings in the study region, and they are concentrated in commercial zones. Besides, a small number of random distributions of factory-type buildings are located in industrial districts.

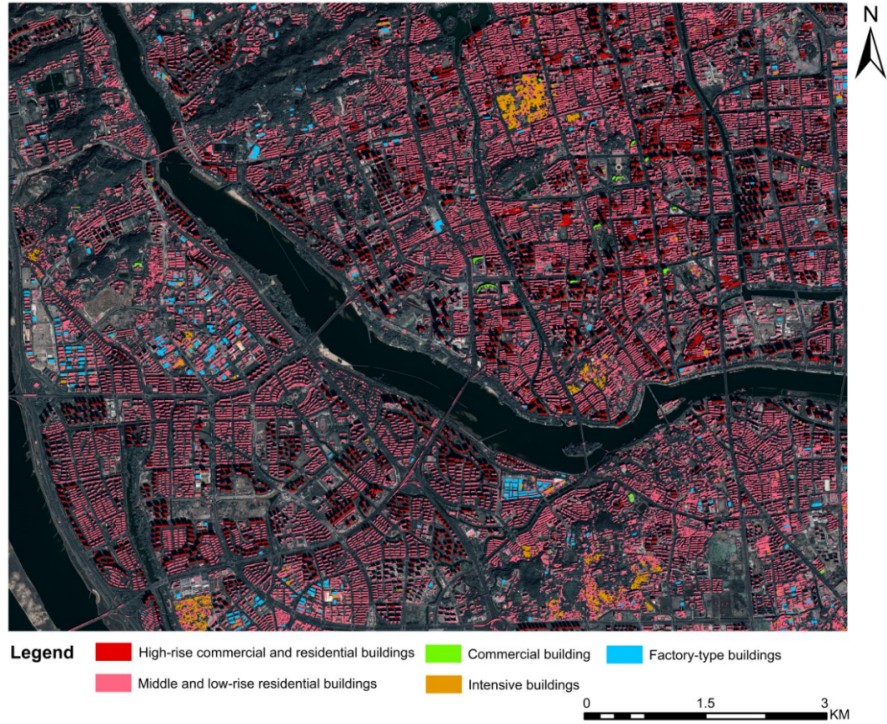

**Figure 10.** Building type classification results of the study area.

### 4.3. Urban Functional Region Classification Results

Figure 11 illustrates the UFR classification results, and Table 5 details the quantitative metrics. Our method yielded an OA of 87.40% on the testing dataset of URF. All UFR categories have achieved accuracy over 75%, which indicates the effectiveness of our classification method. The highest classification accuracy was obtained by residential regions, followed by industrial regions. The most frequent error was the misclassification of 12 commercial regions as residential regions. The mix-use characteristics of high-rise commercial and residential buildings often leads to a confused interpretation of their stated function, which in turn contributes to the misclassification. These results indicate that our method can be beneficial for UFR classification, and in the future we will investigate broader and fine-grained UFR categories, as well as more specific classification rules.

**Table 5.** Confusion matrices of the UFR classification on the study dataset. R1-R4 refer to residential, commercial, industrial and other UFRs. UA and PA refer to user accuracy and producer accuracy.

| Classified | Reference | | | | |
| --- | --- | --- | --- | --- | --- |
| | **R1** | **R2** | **R3** | **R4** | **UA** |
| R1 | 246 | 12 | 4 | 0 | 94.06 |
| R2 | 10 | 38 | 2 | 0 | 78.53 |
| R3 | 8 | 3 | 35 | 0 | 80.00 |
| R4 | 0 | 0 | 0 | 92 | 100 |
| PA | 93.27 | 75.34 | 86.21 | 100 | % |

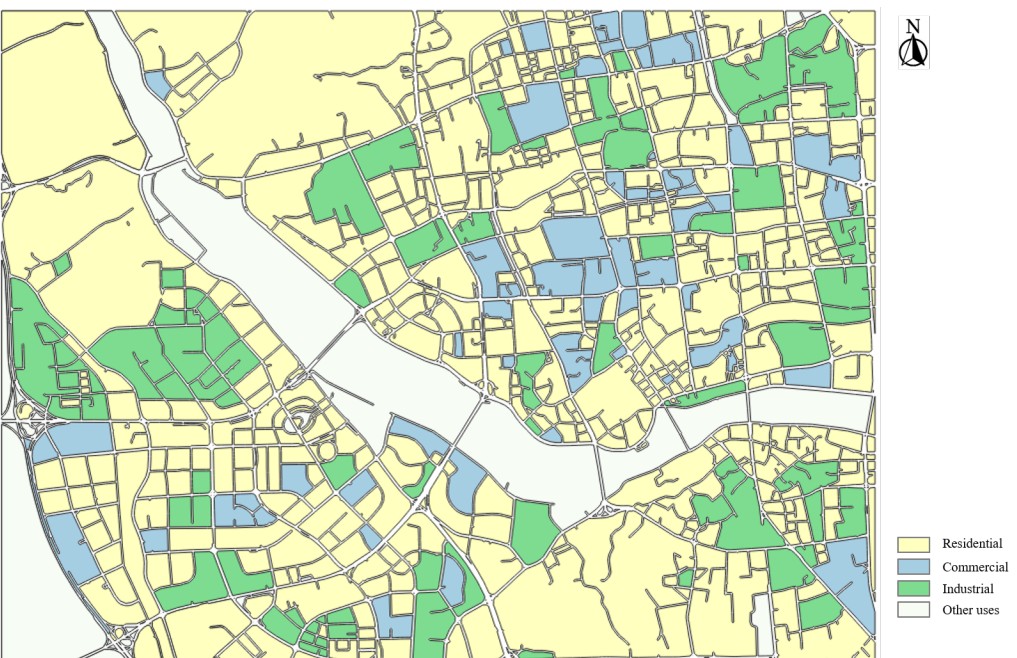

**Figure 11.** Urban functional region (UFR) classification results of the study area.

## 5. Discussion on the Workflow and Analysis Unit

This study investigates to automatically identify urban functional regions (UFRs) from high-resolution remote sensing images. We propose a novel context-aware segmentation network to extract buildings and roads simultaneously. We use a region context module (RCM) and an affinity loss to extract multi-level contextual information and optimize boundary pixels. The proposed network has a high potential to achieve fast and accurate building and road extraction in complex urban environments. Moreover, we investigate to use image-derived road networks to determine basic functional units automatically. Our results showed that the proposed method is effective for UFR classification from high-

resolution remote sensing images in complex urban areas. In contrast to existing studies, the main advantage of the proposed method is that it merely uses high-resolution images to automatically achieve both the determination of functional units and classification of functional regions, rather than using existing boundary datasets, which broadens the use of the proposed method to large-scale UFR mapping.

The proposed UFR classification relies on high-quality remote sensing images. In practice, high-resolution images are often acquired with light shadow and occlusions which may affect building and road extraction, and the subsequent UFR classification. Moreover, the proposed UFR classification depends on five building type results classified from images. The classification accuracy of residential buildings is the highest among many others. This type also accounts for the largest proportion of the study area. By contrast, commercial buildings have the lowest classification accuracy of 52.38% because of high intra-class similarity and inter-class dissimilarity of geometry, height, and morphology. Furthermore, some buildings were obscured by shadows, leading to the fragmented extraction of commercial buildings. Therefore, future studies can be conducted to improve the identification of commercial buildings for the improvement of UFR classification. Moreover, because our functional regions are based on building types, natural features such as green spaces and rivers, may be absent from our function categories. In the future, we will explore the building type classification with finer categories, together with extracting more meaningful features, to further improve UFR classification.

It is also noteworthy that our functional units are determined based upon image-derived road blocks, which have been widely used in urban land use and functional zoning studies. Unlike official road blocks, the image-derived road blocks depend heavily on the results of road extraction. Therefore, the granularity of road blocks may vary between image-derived roads and official roads. Particularly, due to the spatial resolution of used images, the image-derived roads may not be able to identify small roads, resulting in discontinuous roads. These roads are ignored during the post-processing of road extraction. By doing so, some of the obtained functional units may be larger than that of official road networks, failing to separate fine-grained functional regions. In the future, we plan to improve road extraction by using more advanced models, and investigate to fuse UFR results with those based upon other spatial units such as grids and image objects.

## 6. Conclusions

This study proposes an end-to-end method to automatically identify urban functional regions from high-resolution remote sensing images effectively. We tested our method on a GF-2 high-resolution remote sensing image with spatial resolution of 0.8m in Fuzhou, China. The experiment results show that the proposed context-aware network can separate various contextual dependencies by using a region context module and an affinity loss. The proposed network outperforms other existing models with a minimum 1.37% and 1.19% higher F1 score for road and building segmentation. The classification results indicate that the overall accuracy of building type classification is 82.98%, while the accuracy of all four categories of urban functional regions is greater than 75.28%. The results demonstrate the effectiveness of our method. In future studies, we will test our method on a larger scale and consider comparing it with other methods using different urban functional units.

**Author Contributions:** W.Z. (Wufan Zhao) involved in the design of this study, wrote the majority of the paper, set up, and performed the experimental analysis. M.L. conceptualized the aim of this research, wrote, and reviewed this paper. C.W. and G.C. helped with experiments analysis. W.Z. (Wen Zhou) helped with methodology design and manuscript review. All authors have read and agreed to the published version of the manuscript.

**Funding:** This research was funded by the Natural Science Foundation of Fujian Province, China (Grant No. 2021J01630).

**Conflicts of Interest:** The authors declare no conflict of interest.

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
