# Peer review of "Identifying Urban Functional Regions from High-Resolution Satellite Images Using a Context-Aware Segmentation Network"

_remotesensing, doi:10.3390/rs14163996_

Round 1

Reviewer 1 Report

This paper develops an efficient end-to-end sensing building and road segmentation network using high-resolution backbone network for feature extraction and regional context modules to directly predict urban functional region. Affinity loss is used to supervise network training based on regional affinity mapping. Multi-scale images of roads and buildings were analyzed and multi-dimensional features were integrated to infer building types. The overall urban function regions based on the units are defined by the derived road network The experimental results are better than the other 11 convolutional neural network segmentation methods. The classification results show that the overall accuracy of building classification is 82.98%, and the accuracy of urban functional regions is 87.40%.

The overall structure of the paper is logical, hierarchical and accurate. The experimental process is detailed and design is rigorous. The optimization effect of the results is obvious. However, there are still the following problems:

1. Light shadow has a certain influence on the recognition of the edge of buildings. As the point of spectral wave taken by remote sensing satellite image is relatively high, light irradiation will form a certain black shadow area, which will cause the occlusion of nearby road surface or low-rise buildings ;

2. Consider whether the classification method is reasonable. The current building type is mainly divided by the height of the building. In actual production and life, it is difficult to completely distinguish between the commercial buildings and residential buildings. The division method described in the article can not make the division of high-rise residential area ;

3. There are problems in road identification, and many discontinuous lines appear in the segmentation results of low-level roads. It is suggested to consider whether to specify the width and type of roads to be divided, and how to deal with small roads such as alleys, community streets and sidewalks.

4. The paragraphs on lines 248 and 257 need to be indented, and the graphic pages on pages 10 and 16 have too much white space.

In view of the classification and segmentation of experimental results, it is suggested to refer to the following articles about remote sensing :

Xu Sheng, Zhou Kai, Sun Yuan and Yun Ting(*). "Separation of Wood and Foliage for Trees From Ground Point Clouds Using a Novel Least-Cost Path Model." in IEEE Journal of Selected Topics in Applied Earth Observations and Remote Sensing, vol. 14, pp. 6414-6425, 2021, doi: 10.1109/JSTARS.2021.3090502.

Xu Sheng, Xin Li, Jiayan Yun and Shanshan Xu(*). "An Effectively Dynamic Path Optimization Approach for the Tree Skeleton Extraction from Portable Laser Scanning Point Clouds." in Remote Sensing. vol. 14(1), pp. 94, 2022, doi: 10.3390/rs14010094

 Xu Sheng, Zhou Xuan, Ye Weidu and Ye Qiaolin(*). "Classification of 3D Point Clouds By A New Augmentation Convolutional Neural Network." IEEE Geoscience and Remote Sensing Letters, vol. xx, pp. 1-5, 2022, doi: 10.1109/LGRS.2022.3141073

Sun C, Yun T*, Individual Tree Crown Segmentation and Crown Width Extraction From a Heightmap Derived From Aerial Laser Scanning Data Using a Deep Learning Framework. Front. Plant Sci. 2022. 13:914974. doi: 10.3389/fpls.2022.914974.

Chen, X., Yun, T*, "Individual Tree Crown Segmentation Directly from UAV-Borne LiDAR Data Using the PointNet of Deep Learning [J]," Forests. 2021. 12(2): 131.

Reviewer 2 Report

This paper proposes a deep learning based urban functional region identification method from remote sensing imagery. The idea in the this paper is interesting and the contribution of this paper is well addressed. There are still some points needed to be specified clearly and solved. 

1.        It is recommended to use clearer figures to illustrate the pipeline of the method. For example, the notations in Figure 4 should correspond to the text in 3.3.1, and the abbreviations in figures like FSConv should be introduced in the text.

2.        In line 200, I suppose it should be “c. Affinity loss” rather than “b. Affinity loss”.

3.        More details about the matching processing of buildings and their shadows should be provided.

4.        The evaluation indexes are not corresponding in 4.1.1 and Table 3. In the text, they are mIoU and OA, while in table 3, they are OA and F1.

5.        It is better to use some instances (image samples) to support the conclusion in lines 368-376.

6.        More experimental datasets and results with large area results should be provided. Moreover, it would be better to compare the proposed method with other recently published UFZ detection approaches in the RS field shown in [1] and [2].

[1]     Lu, W., Tao, C., Li, H., Qi, J., Li, Y., 2022. A unified deep learning framework for urban functional zone extraction based on multi-source heterogeneous data. Remote Sens. Environ. 270, 112830.

[2]     Du, Shouhang, Du, Shihong, Liu, B., Zhang, X., 2021. Mapping large-scale and fine-grained urban functional zones from VHR images using a multi-scale semantic segmentation network and object based approach. Remote Sens. Environ. 261, 112480.

Reviewer 3 Report

The authors of the manuscript# remotesensing-1803951 aim to automatically classify urban functional regions using high-resolution remote sensing data and a context-aware segmentation network. The novelty of their work is to extract buildings and roads, which often require implementing counter-active detection philosophies, such as detection of connectedness of roads vs. discreet clusters of building types. After reading the manuscript, this reviewer suggests the paper may be considered for publication after the authors have addressed the minor corrections described below and any suggestions and concerns of the other reviewers and the editor(s).

Please find my comments below:

  1. The authors show that their method yields higher accuracy than other CNN-based segmentation methods. One aspect of deep learning algorithms is their need for extensive input data and preprocessing. The reviewer thinks for a comprehensive comparison among available methods, the authors need to discuss this aspect of their work in the manuscript. E.g., does the proposed method require more, less, or similar data volume compared to other CNN-based segmentation methods, are there any extra preprocessing required compared to other methods? If yes, the author may justify how such additional data and computational needs are justified by increased accuracy.
  2. Lines 39-41: Please provide a reference for NYCDCP in the reference list.
  3. Figure 8: The bar labels may be changed from a comma (,) separator to a dot separator (.) to indicate decimal places (e.g., 52,38 -> 52.38).

Round 2

Reviewer 2 Report

The authors have kindly addressed all of my concerns. I am glad to recommend the publication of this paper.